# The Impact of Seller Trust in a C2C Platform on Golf Club Purchase Intention and the Interaction Effect of Regulatory Focus

**DOI:** 10.3390/bs14060479

**Published:** 2024-06-06

**Authors:** Chulhwan Choi, Inyup Lee, Hosuk Yoo

**Affiliations:** 1Department of Physical Education, Gachon University, 1342, Seongnam-daero, Sujeong-gu, Seongnam-si 13120, Republic of Korea; chulhwanchoi@gachon.ac.kr; 2Department of Sport Science, Jeju National University, 102, Jejudaehak-ro, Jeju-si 63243, Republic of Korea; 3Graduate School of Physical Education, Kyung Hee University, 1732, Deokyeong-daero, Giheung-gu, Yongin-si 17104, Republic of Korea

**Keywords:** C2C market, seller trust, regulatory focus, purchase intentions, used goods

## Abstract

(1) Background: This study aimed to investigate the influence of seller trust on the purchase intention of consumers of used golf clubs in the context of increasing C2C transactions and further explore the interaction effect of regulatory focus. (2) Methods: Data were collected from 200 participants who had experience purchasing golf clubs through the Carrot Market platform, employing a 2 × 2 experimental design. An independent samples *t*-test was utilized to examine the effect of seller trust on purchase intention, followed by a two-way analysis of variance to assess the interaction effect of regulatory focus. (3) Results: The results revealed a significant difference in purchase intention based on seller trust, with higher purchase intentions observed when seller trust was high compared to when it was low. Additionally, the interaction effect of regulatory focus was found to be significant in the impact of seller trust on golf club purchase intention. Specifically, when seller trust was high, no significant differences were observed among control focus groups. However, when seller trust was low, promotion-focused consumers exhibited higher purchase intentions than prevention-focused consumers. (4) Conclusions: These findings underscore the importance of seller trust in the context of an expanding market for online second-hand trading platforms.

## 1. Introduction

The advent of the digital era has brought about fundamental changes in consumer purchasing and selling behaviors, accompanied by the emergence of the Consumer to Consumer (C2C) business model [1]. At the core of these changes lies online intermediary platforms, which have shifted away from traditional enterprise-centric distribution structures, enabling consumers to directly fulfill the roles of sellers and buyers [2]. This structural transformation in distribution can be observed through the growth trajectory of the second-hand goods market. As exemplified by a study conducted by Statista [3], the second-hand goods market in the United States alone is projected to reach USD 64 billion in 2024, with the domestic market showing rapid growth, as evidenced by the trading volume of platforms like “Carrot Market” surpassing KRW 1 trillion as of 2022 [4].

The driving force behind the growth of second-hand trading can be attributed significantly to the impact of the COVID-19 pandemic. As the economic downturn persists, consumers have shown a preference for affordable second-hand transactions over purchasing high-value items, leading to widespread adoption of mobile platforms for such transactions and a shift in consumer perception toward second-hand goods. Of note is the surge in popularity of the golf market since the spread of COVID-19. With an increasing number of Millennials and Generation Z individuals taking up golf, the market has seen a trend toward second-hand trading for golf equipment and apparel, driven by the relatively high prices and diverse range of golf products, thus alleviating consumer financial burdens. Consequently, second-hand trading can be regarded as an inevitable phenomenon arising from both the growth of national economies and changes in consumer behavior, further accelerated by significant investments from major retail conglomerates.

In the past, consumers had to consider temporal and economic costs when selling products, often resorting to offline or online markets to find suitable middlemen. However, the emergence of second-hand trading platforms, serving as intermediaries connecting supply and demand, has enabled consumers to make convenient and efficient decisions [5].

Nevertheless, despite the rapid growth of the second-hand trading market, it is not without its challenges. Consumers make decisions based on trust due to the high uncertainty and complexity of the market [6]. According to Park and Cheon [7], information provision in C2C transactions relies solely on the seller, resulting in a lack of reliable information regarding the condition, functionality, and warranty periods of products. Additionally, given the anonymous nature of sellers on second-hand trading platforms [8], the complexity is exacerbated by the rise in double-dealing issues, leading to frequent occurrences of consumer losses, especially with the increase in non-professional sellers [5]. This study will explore the mechanisms through which trust can be better established and maintained, potentially leading to more secure and satisfactory transactions for all parties involved. The findings of this research will not only enrich theoretical discussions around trust and consumer behavior in digital marketplaces but also provide practical guidance for platform developers and policymakers aiming to enhance the efficacy and reliability of C2C platforms.

Transactions between individuals in an online setting are inherently limited in terms of information about the other party. Thus, seller trust can significantly influence consumer purchasing behavior. The issue of seller trust affects consumer purchase decisions, leading to consumer losses due to information asymmetry and the risk of fraudulent transactions [5]. Montague and Asan [9] argued that trust plays a critical role in consumer-to-consumer trading platforms involving sellers, platform operators, and relationships between consumers, while Um et al. [10] emphasized the centrality of trust in consumer purchasing decisions in online transactions. Survey results on consumer inconveniences in second-hand trading from a buyer’s perspective indicate concerns about the quality and condition of goods (47%), apprehensions about fraudulent transactions (46%), difficulties in contacting sellers and negotiating prices (33%), and concerns about personal information exposure (22%) [11].

Meanwhile, the regulatory focus theory within prospect theory explains how individuals regulate and control their behavior to achieve specific goals [12]. According to Crowe and Higgins [13], regulatory focus entails two psychological behavioral types that people exhibit to achieve their goals, namely promotion focus and prevention focus. Promotion-focused individuals focus on achieving goals, while prevention-focused individuals strive for stable prevention by minimizing losses [14]. Previous studies on regulatory focus reveal its influence on consumer perceptions of problems, information processing, memory, and evaluations of alternatives in consumer decision-making [13,15]. By incorporating these dimensions of regulatory focus, the present research will offer new insights into how different motivational orientations can further complicate or facilitate trust-building processes in C2C transactions, thereby impacting purchase intentions in a significant way.

This study’s findings will not only enrich theoretical discussions around trust and consumer behavior in digital marketplaces but also provide practical guidance for platform developers and policymakers aiming to enhance the efficacy and reliability of C2C platforms.

### 1.1. Theoretical Background

#### 1.1.1. Seller Trust in C2C Platforms

Consumer-to-Consumer (C2C) platforms, such as online marketplaces, facilitate transactions between numerous sellers and buyers, making the trust in sellers a paramount factor for successful exchanges [16]. In these platforms, where transactions occur between individuals rather than through traditional corporate structures, trust is crucial because it directly influences consumer purchase decisions [17]. Trust in this context stems from the reliability of the information provided by sellers, which is not overseen by any large corporate entity but by the individuals themselves.

In the C2C environment, theories such as trust transfer theory explain how trust established in one context, like a trusted platform, extends to another, such as the sellers using that platform [18]. For instance, a study by Zhao, Huang, and Su [19] showed that simultaneous trust in sellers and the brands they represent enhances consumers’ continuous purchase intentions. They found that informational and emotional support from sellers can fortify this trust, significantly influencing purchase decisions. Social exchange theory also highlights the role of social interactions in building trust. This theory posits that positive social exchanges, such as user reviews and direct interactions between buyers and sellers, enhance trust and thereby reduce the perceived risk associated with transactions [20]. Such dynamics are crucial in C2C platforms where formal guarantees are often less apparent than in traditional retail settings.

Additionally, the theory of reasoned action suggests that consumers’ attitudes toward the trust of a seller directly influence their purchasing intentions [21]. Positive perceptions of a seller’s trust, therefore, lead to higher likelihoods of transaction completions. Empirical research supports these theories by showing how various elements of trust interactions affect consumer behavior [22]. For example, Chen et al. [18] examined how trust in a platform positively affects trust in individual sellers and how this, in turn, impacts purchase intentions. Their findings also highlight the importance of perceived effectiveness of e-commerce institutional mechanisms and perceived website quality as key factors that moderate the relationship between platform trust and seller trust.

Moreover, the study by Li et al. [23] using social network and reputation theories demonstrates how trust and recommendations within C2C platforms mimic real-life interpersonal trust dynamics, significantly impacting online consumer behavior. Furthermore, Hasim et al. [24] and Anantharaman et al. [25] have both explored the influence of personal characteristics and social dynamics on trust. They found that the expertise and trustworthiness of streamers in live-streaming sales platforms significantly boost consumer purchase intentions, underscoring the importance of personal traits in trust development. Social integration and the bandwagon effect on social commerce platforms are shown to play crucial roles in enhancing consumer trust and purchase intentions, highlighting the significant influence of social interactions and public opinions.

These findings collectively suggest that both individual and social factors are pivotal in shaping trust and influencing consumer decisions on C2C platforms, leading to the proposal of the following hypothesis: Trust in individual sellers on C2C platforms, moderated by factors like social dynamics and institutional mechanisms, decisively influences consumer purchase intentions.

**H1.** 
*High seller trust in C2C platforms will result in higher purchase intentions compared to low seller trust.*


#### 1.1.2. Regulatory Focus Theory

Regulatory focus theory, developed by Higgins [26], distinguishes two distinct motivational orientations: promotion focus and prevention focus. Individuals with a promotion focus are driven by the pursuit of positive outcomes and motivated by desires for achievement, advancement, and progress. In contrast, prevention focus involves a preference for avoiding negative outcomes, emphasizing safety and security. These orientations are not static but fluctuate with situational factors, reflecting the dynamic and context-dependent nature of regulatory focus.

The implications of these dynamics are particularly evident in digital environments, where the way consumers interact with online content and platforms can vary greatly depending on their regulatory focus. For example, the literature suggests that promotion-focused individuals are more receptive to marketing that highlights potential gains or positive outcomes associated with a product or service, whereas prevention-focused individuals are more influenced by marketing that emphasizes security, risk reduction, and reliability [27,28].

Additionally, the privacy paradox in social media has been explored, illustrating how promotion and prevention-focused behaviors guide decisions to share or protect personal information, applying the regulatory focus theory to understand these decisions. The impact of social media marketing on consumer responses has further detailed how traditional regulatory focus dynamics are modified due to the interactive and dynamic nature of these platforms. Furthermore, the profound effects of digital and social media marketing on consumer behavior have been emphasized, demonstrating how these marketing strategies can influence regulatory focus and impact e-commerce decision-making processes.

Moreover, studies indicate that the regulatory focus influences consumers’ perceptions of trust and risk in online settings. Promotion-focused consumers have a higher tendency to trust online platforms and perceive lower risks, which potentially increases their willingness to make purchases. Conversely, prevention-focused consumers require more robust trust signals to overcome their higher perceived risks, affecting their purchasing behavior on C2C platforms [29].

Building on earlier findings, researchers such as Higgins and Spiegel [30], Jeong et al. [31], and Bryant and Dunford [32] provide further evidence that promotion-focused individuals are sensitive to gains, while prevention-focused individuals are more responsive to potential losses, influencing their information processing and decision-making under risk.

Previous studies by Lee et al. [33] and Liberman et al. [34] have also investigated how information presentation and product innovation relate to regulatory focus, revealing that promotion-focused individuals have higher purchase intentions under positive effect messages, while prevention-focused consumers show similar intentions under conditions of high product performance confidence.

Given these insights, we propose the following hypothesis (Figure 1): This hypothesis suggests that understanding and leveraging the regulatory focus of consumers can significantly modulate their response to different marketing approaches, emphasizing the need for tailored marketing strategies that resonate with the specific motivational orientations of the target audience.

**H2.** 
*The interaction effect of regulatory focus (promotion vs. prevention), influenced by the context of the transaction, on the impact of seller trust on purchase intentions in C2C platforms will be significant.*


**H2-1.** 
*In contexts where seller trust is high, and the situation emphasizes safety and reliability, prevention-focused consumers will exhibit higher purchase intentions.*


**H2-2.** 
*In contexts where seller trust is low, but the opportunity for gains is emphasized, promotion-focused consumers will exhibit higher purchase intentions.*


## 2. Research Methodology

### 2.1. Research Design and Participants

This study aimed to examine the influence of seller trust (high/low) on purchase intention for golf equipment on C2C platforms and further compare purchase intentions based on individual regulatory focus. Thus, a 2 (seller trust) × 2 (regulatory focus) design was employed. The detailed design for conducting the research is as follows:

Based on the survey results from Consumerinsight [11], Carrot Market (87%), Jungonara (26%), and Bungaejangteo (20%) were identified as the top C2C platforms in the Republic of Korea. Carrot Market was selected as the C2C platform for this study. Subsequently, to select the product category for golf equipment, a focus group interview was conducted with 20 individuals who had experience purchasing used golf equipment on Carrot Market. When asked about their interest in golf equipment and the products they purchased, 15 out of 20 individuals mentioned golf drivers. Therefore, golf drivers were selected as the product category for the experimental stimuli.

The participants for this study were recruited from individuals who had experience with purchasing golf equipment through Carrot Market. Using the G-power program, the minimum sample size required for this study was determined. As a result, 76 participants were needed (f = 0.25, α = 0.05, Power = 0.95, groups = 4, measurements = 2). To clearly distinguish between regulatory focus groups, a total of 200 participants were recruited using convenience sampling methods. Subsequently, 140 valid samples were utilized for testing hypothesis 2 after segregating the participants based on their regulatory focus. The participant pool was predominantly male, accounting for 60.5% (121 individuals), with females comprising 39.5% (79 individuals). The age of participants varied, with the largest group being those aged 20 to 29 years (43.0%, 86 individuals), followed by 30 to 39 years (36.0%, 72 individuals), 40 to 59 years (13.0%, 26 individuals), and those over 50 years old (8.0%, 16 individuals). In terms of purchase frequency through the app, participants reported a range of experiences: 1–3 purchases (9.0%, 18 individuals), 4–7 purchases (22.0%, 44 individuals), 8–10 purchases (28.5%, 57 individuals), and more than 10 purchases (40.5%, 81 individuals). The golfing experience among the participants also varied: less than 6 months (18.5%, 37 individuals), 6 months to less than 1 year (39.5%, 79 individuals), 1 to less than 3 years (31.0%, 62 individuals), and over 3 years (11.0%, 22 individuals). The study was designed to ensure anonymity and to collect only non-sensitive demographic information, thus minimizing any risk to the participants. This approach was adopted to adhere to ethical standards for human subject research that might not require formal Institutional Review Board approval due to the minimal risk involved.

### 2.2. Stimulus Selection and Research Instruments

(1)Experimental Stimulus Development

Virtual seller accounts and manner temperatures were manipulated to present information about golf drivers in a simulated Carrot Market platform scenario. Online transactions are characterized by trust-based transactions [35]. Specifically, sellers possess more information about the products than buyers due to their experience using the products, leading to information asymmetry between sellers and buyers [36]. Consequently, seller trust is crucial in the transaction process since buyers make purchase decisions solely based on the information provided by sellers. Carrot Market utilizes a composite indicator called ‘manner temperature’, derived from compliments, reviews, manner evaluations, and operator sanctions from buyers, to help buyers easily assess seller trust. Therefore, in this study, seller trust in Carrot Market was manipulated as high (50.5 °C) and low (30 °C). Additionally, during used transactions, sellers provide various information to prospective buyers (e.g., purchase price, quality condition, usage period, performance), and the seller’s price for the new product is often presented as well [8]. This information is provided to imply that the selling price is fair due to the seller’s high purchase price for the product. Therefore, in addition to the information provided by sellers to facilitate transaction completion, this study also presented textual information related to the purchase price, quality condition, and product performance of the selected product category, along with an image of the product.

The experimental stimuli were all designed to resemble the layout, buttons, and colors of the Carrot Market platform (see Figure 2). Virtual brands were used, and participants were randomly assigned to the experimental stimuli with manipulated seller trust on Carrot Market to avoid bias resulting from pre-existing consumer attitudes toward real brands and product categories.

### 2.3. Research Instruments

To measure the variables used in the study, an online survey was conducted using a questionnaire developed by Google, with modifications and enhancements made based on items used in previous research related to the variables. First, a pre-question was used to filter respondents based on their experience with golf equipment transactions on Carrot Market. Subsequently, to measure regulatory focus, ten items were created, consisting of five promotion focus items and five prevention focus items, based on previous studies [37,38]. Next, to measure the dependent variable, purchase intention, three items were presented, modified, and enhanced to fit the context of this study based on previous research by Mackenzie et al. [39] and Stafford et al. [40]. Finally, seventeen items were included in the questionnaire to ascertain respondents’ general characteristics (gender, age, frequency of Carrot Market usage, golfing experience). All items except demographic questions were measured on a 5-point Likert scale (1 = Strongly Disagree, 5 = Strongly Agree).

The online survey randomly sent different experimental stimuli, based on the manipulation of seller trust (manner temperature: 50.5 °C/30 °C) on Carrot Market, to 200 participants via smartphones for self-administered responses, which were then utilized as valid data.

### 2.4. Manipulation Check of Experimental Stimuli

Prior to the survey, a pilot test was conducted with 20 participants to ensure that the respondents perceived the manipulation of seller trustworthiness on Carrot Market (50.5 °C/30 °C) as intended by the researcher. For this manipulation check, a statement was used: “Based on the manner temperature of the seller, the seller seems trustworthy”. An independent samples *t*-test was conducted for manipulation check, and the results are presented in Table 1. The results of this manipulation check confirmed that the experimental stimuli were manipulated as intended by the researcher.

### 2.5. Validity and Reliability Verification of Measurement Instruments 

Before the analysis, content validity and construct validity verification procedures were conducted for the measurement instruments used in this study. First, content validity was confirmed through a step-by-step process involving validation by two professors specializing in the sports industry and one doctoral-level researcher. Second, confirmatory factor analysis was conducted to verify construct validity. The results of confirmatory factor analysis and the fit indices of the measurement model satisfied the goodness-of-fit criteria proposed by Browne and Cudeck [41] (CFI > 0.90, IFI > 0.90, TLI > 0.90, RMSEA < 0.08), as shown in Table 2.

## 3. Results

### 3.1. Descriptive Statistics Analysis

In this study, independent samples *t*-test and two-way analysis of variance (ANOVA) were conducted to examine the interaction effect of regulatory focus on the impact of C2C platform seller trust on purchase intention for golf equipment. Before conducting the two-way ANOVA, the normality and homoscedasticity of the data were checked to ensure that the data met the assumptions for ANOVA [42]. First, skewness and kurtosis were examined to assess normality. The skewness ranged from −0.21 to −0.58, and the kurtosis ranged from −0.29 to −0.38. These results fell within the acceptable ranges proposed by West et al. [43] for skewness (±2) and kurtosis (±7), indicating that normality was met.

### 3.2. Hypothesis Testing

(1)Impact of C2C Platform Seller Trust on Purchase Intention for Golf Equipment

An independent samples *t*-test was conducted to examine the first hypothesis, which states that the trust of C2C platform sellers influences the purchase intention for golf equipment. The results presented in Table 3 indicate a statistically significant difference in purchase intention based on the seller’s trust. Specifically, participants showed higher purchase intention when seller trust was high compared to when it was low (F = 8.35, *p* < 0.001). Thus, Hypothesis 1, which posits that higher seller trust leads to higher purchase intention, was supported.

(2)Interaction Effect of Regulatory Focus on the Impact of C2C Platform Seller Trust

To test the interaction effect of regulatory focus on the impact of C2C platform seller trust on purchase intention for golf equipment, a two-way ANOVA was conducted. The results presented in Table 4 indicate a statistically significant interaction effect between seller trust and regulatory focus on purchase intention (F = 14.09, *p* < 0.001).

Further analysis of the interaction effect was visualized in Figure 3. The figure demonstrates that while there was no significant difference in purchase intention between promotion-focused and prevention-focused consumers when seller trust was high (*t* = −0.00, *p* > 0.05), there was a significant difference when seller trust was low (*t* = 4.23, *p* < 0.001). Specifically, consumers with a promotion focus showed higher purchase intention than those with a prevention focus when seller trust was low. Based on these results, Hypothesis 2-1 was rejected, and Hypothesis 2-2 was supported.

## 4. Discussion

### 4.1. Influence of C2C Platform Seller Trust on Purchase Intention for Golf Equipment

The results confirmed that higher seller trust on C2C platforms leads to higher purchase intentions for golf equipment compared to situations where trust is low. Consumers base their decisions on trust due to the high uncertainty and complexity of transactions in the market [6]. Trust not only reduces economic transaction costs but also cognitive transaction costs such as risk, uncertainty, and complexity, thereby increasing consumer satisfaction and promoting consumer behavior [44,45]. Particularly in peer-to-peer transactions, where consumers rely on subjective and general information during product exploration and transactions, the results of this study are understandable. Moreover, platforms like “Carrot Market” provide features where trust in sellers can be identified and selected from a buyer’s perspective through the recording of transaction history, evaluations, and reviews, which enables consumers to have confidence in the product’s condition.

AlSheikh, Shaalan, and Meziane [46] highlight the importance of trust in overcoming the challenges posed by the digital nature of C2C transactions, which lack physical interaction and direct verification. Similarly, Wei, Li, Zha, and Ma [16] discuss how institution-based trust influences transaction intentions, advocating for stringent verification processes to foster trust among platform users. These findings are supported by Chen et al. [18] and Pan et al. [47], who emphasize the positive impact of trust on purchase intentions in C2C online platforms.

From a practical perspective, the development of peer-to-peer trading platforms should prioritize introducing authentication systems where sellers undergo detailed inspections of specific information such as the initial price, description, and photos of the product to ensure price adequacy and accuracy of condition [18,47]. Only accounts designated as ‘certified sellers’ or ‘certified products’ after verification by administrators should be allowed to sell used items. Furthermore, implementing data mining technology within the platform to distinguish between normal and fraudulent posts and providing visual data to users can significantly enhance trust and security, thereby improving consumer confidence in the platform and the products offered.

### 4.2. Interaction Effect of Regulatory Focus on the Impact of C2C Platform Seller Trust

The second hypothesis of this study investigated the interaction effect of regulatory focus on the impact of C2C platform seller trust on purchase intention for golf equipment. The statistical analysis revealed a significant interaction effect. Specifically, when seller trust was high, there was no significant difference in purchase intention between promotion-focused and prevention-focused consumers. However, when seller trust was low, promotion-focused consumers showed higher purchase intention than prevention-focused consumers.

This finding aligns with research by Kim and Park [48] and Ross [49], which demonstrates how regulatory focus influences consumer attitudes and behaviors toward various types of purchases. These studies highlight that promotion-focused consumers are more likely to engage in purchasing behaviors when trust cues are strong, reflecting their sensitivity to positive information and rewards. Conversely, prevention-focused consumers require high levels of trust to mitigate their inherent risk aversion, making them more cautious.

Further research by Partouche et al. [50] found that millennials respond more positively to CRM marketing strategies that are aligned with their regulatory focus, especially under conditions that match their inherent inclinations toward promotion or prevention. These strategies, when tailored to the regulatory focus of the consumers, can significantly enhance the effectiveness of marketing efforts. 

Additionally, Werth and Foerster [27] provide further insight by demonstrating that consumers’ regulatory focus significantly influences their product evaluations and decision-making processes. Their studies suggest that while promotion-focused consumers prioritize aspirational features in products, prevention-focused individuals focus on safety and security features. This difference in focus can be leveraged by marketers to customize their messages and product offerings on C2C platforms to better meet the distinct needs of each consumer group [27].

In conclusion, this study’s results support the basic assumptions of regulatory focus theory, suggesting that individual motivations for goal achievement can be categorized into promotion and prevention focuses, which influence consumer purchasing decisions. When seller trust is high, all consumers show similar levels of purchase intention, possibly due to the universally positive impact of high trust. However, when seller trust is low, promotion-focused consumers exhibit higher purchase intentions, likely due to their willingness to take risks and pursue gains. Platforms like “Carrot Market” and “KREAM” adapt their operations to cater to prevention-focused consumers, thus reducing their apprehensions and encouraging transactions. Therefore, there is a need to tailor marketing strategies to different segments of potential consumers based on their regulatory focus.

## 5. Conclusions and Recommendations

This study extends previous research on C2C platforms by examining the influence of seller trust and the interaction effect of consumers’ regulatory focus on their purchasing behaviors. Notably, whereas past studies predominantly focused on external factors such as product characteristics and pricing, this investigation emphasizes the intrinsic attribute of regulatory focus tendencies in consumers. It elucidates how these tendencies modulate the impact of seller trust on purchasing intentions, offering a nuanced understanding of buyer behavior in second-hand trade contexts.

The research highlights the importance of the relationship between seller trust and individual regulatory focus tendencies in understanding contemporary consumer behaviors in C2C environments. In these platforms, where transactions are directly mediated between buyers and sellers, trust plays a crucial role in shaping purchase decisions. Regulatory focus acts as a lens through which consumers process information and make decisions; thus, understanding the interaction between seller trust and regulatory focus can provide deeper insights into consumer behaviors.

Distinctively, the study showcases how consumer behavior on C2C platforms differs from large-scale online retailing or traditional retailing, attributing these differences to the personal nature of transactions in second-hand markets. The findings suggest that in scenarios where seller trust is low, consumers with a promotion focus exhibit higher purchase intentions compared to those with a prevention focus, underscoring a variance in response based on individual predispositions. Practically, the findings can directly impact marketing strategies in the field. For instance, platforms can leverage these insights to enhance seller–buyer relationships by promoting highly trusted sellers or by customizing advertisements for consumers with specific regulatory focuses. Such strategies can improve platform competitiveness and enhance consumer experiences.

In this study, several limitations have been identified that could impact the generalizability and applicability of the findings. First, the demographic scope of this study was restricted primarily to a specific population within the Republic of Korea. Future research could enhance external validity by including a broader and more diverse participant pool from various regions. Second, the study utilized a cross-sectional design, capturing data at a single point in time and not accounting for changes in consumer behavior or trust dynamics over time. Longitudinal research designs are recommended for future studies to better understand how these variables evolve. Third, the study predominantly focused on two types of regulatory focuses—promotion and prevention. Including a wider range of psychological and motivational factors could provide deeper insights into the complex decision-making processes of consumers in C2C transactions.

To address these limitations and further enrich the understanding of consumer behavior in C2C markets, several recommendations are proposed for future research. Broadening the demographic inclusion to cover different socioeconomic statuses, ages, and cultural backgrounds would validate the findings across different consumer segments. Additionally, integrating more psychological constructs such as risk aversion and consumer expertise could uncover deeper dynamics in consumer interactions on C2C platforms. Employing longitudinal designs would allow researchers to observe the stability of effects and their long-term implications, providing a dynamic view of the interactions between seller trust and consumer behavior. Moreover, conducting field experiments involving actual transactions on C2C platforms could help in understanding the practical implications of seller trust and regulatory focus under real-world conditions. Finally, C2C platforms are encouraged to invest in advanced technological solutions to enhance trust verification, such as AI-driven tools for fraud detection and automated systems for assessing the reliability of sellers, which could significantly improve user experience and platform reliability.

## Figures and Tables

**Figure 1 behavsci-14-00479-f001:**
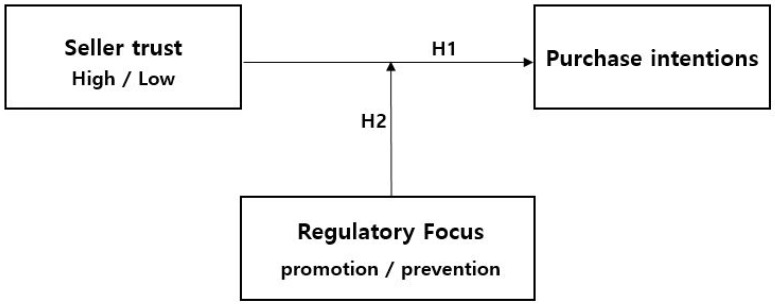
Research model of consumer behavior in C2C platforms.

**Figure 2 behavsci-14-00479-f002:**
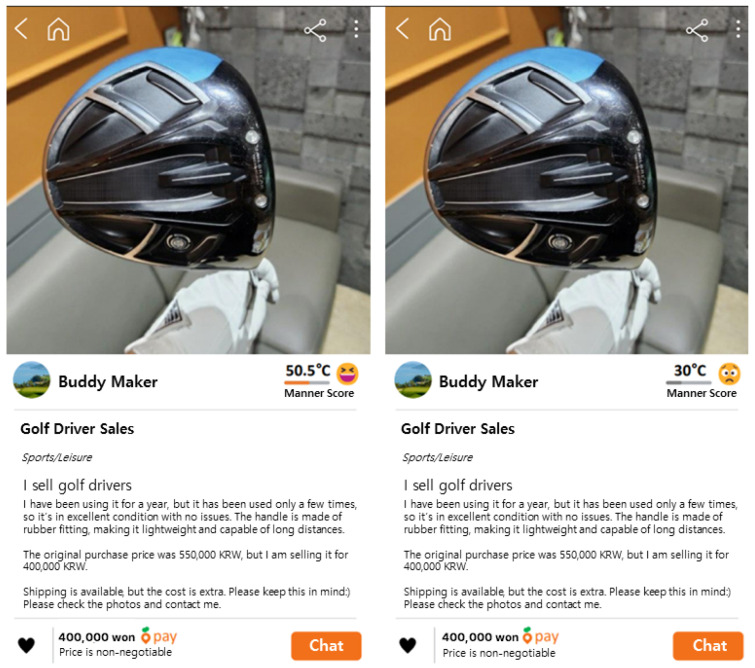
Experiment stimulant.

**Figure 3 behavsci-14-00479-f003:**
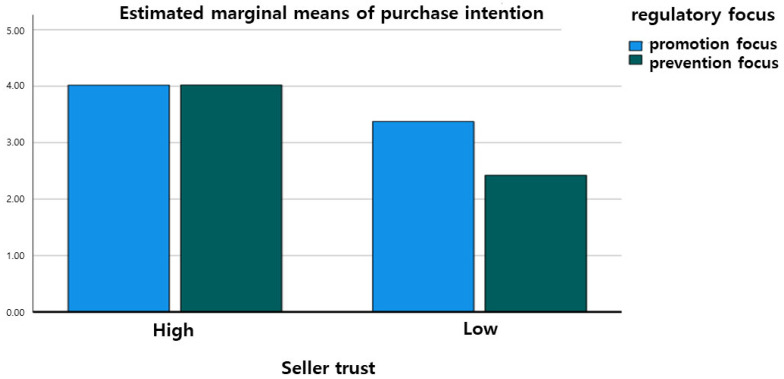
Interaction effect graph.

**Table 1 behavsci-14-00479-t001:** The independent samples *t*-test.

Seller Trust	M	SD	*t*
High	3.10	0.99	2.426 *
Low	4.20	1.03	

* *p* < 0.001.

**Table 2 behavsci-14-00479-t002:** Confirmatory factor analysis.

Factors		Questionnaire			S. C.	S. E.	C. R	*α*	AVE	CR
Regulatory Focus	Promotion	When purchasing a product, if there is something I like about it, I tend to choose it even if I like it.			0.74	0.07	12.80	0.92	0.69	0.91
		When purchasing a product, think about how much you will like it after purchasing it.			0.73	0.06	12.51			
		When purchasing a product, focus on how much you like it.			0.88	0.05	17.07			
		When purchasing a product, I think a lot about its positive aspects.			0.93	0.06	18.77			
		When purchasing a product, I tend to think of the positive aspects of the product first.			0.86	-	-			
	Prevention	If there is something that bothers me when purchasing a product, I tend not to purchase it, even if there are aspects of it that I like.			0.71	0.04	14.01	0.95	0.78	0.94
		When I purchase a product, I tend to think that I might regret it after purchasing it.			0.75	0.05	15.71			
		Focus on how few things bother you when purchasing a product.			0.98	0.02	50.52			
		When purchasing a product, I think a lot about its negative aspects.			0.99	0.01	55.33			
		When purchasing a product, I tend to think of the negative aspects of the product first.			0.97	-	-			
Purchase intention		I am likely to purchase the product in the presented post.			0.80	0.08	11.25	0.85	0.65	0.84
		I will purchase the product in the suggested post.			0.81	0.08	11.32			
		I would like to purchase the product in the suggested post if I get the chance.			0.81	-	-			
Model fit		*X* ^2^	*df*	*p*	IFI	TLI	CFI		RMSEA	
		114.69	60	0.001	0.98	0.97	0.98		0.06	

**Table 3 behavsci-14-00479-t003:** H1 *t*-test.

Seller Trust	*M*	*SD*	Levene’s Test		*t*	*df*	*p*
			F	p			
High	4.06	0.58	8.35	0.004	6.46	198	0.001
Low	3.40	0.84					

**Table 4 behavsci-14-00479-t004:** H2 two-way ANOVA.

Variables	Type III SS	*df*	*MS*	*F*	*p*
Modified model	61.84 a	3	20.1	36.75	0.001
Intercept	1671.46	1	1671.46	2979.78	0.001
Seller trust (A)	43.85	1	43.85	78.17	0.001
Regulatory focus (B)	7.86	1	7.86	14.02	0.001
AxB	7.90	1	7.90	14.09	0.001
Error	76.28	136	0.56		
Total	1799.88	140			
Modified total	138.136	139			

a. R^2^ = 0.44 (modified R^2^ = −0.43).

## Data Availability

Data are contained within the article.

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
