# Peer review of "The Impact of Seller Trust in a C2C Platform on Golf Club Purchase Intention and the Interaction Effect of Regulatory Focus"

_behavsci, 2024, doi:10.3390/bs14060479_

Round 1

Reviewer 1 Report

Comments and Suggestions for Authors

Aim(s) of the research

The authors state the aims of their study as follows: “This study aimed to investigate the influence of seller trust on the purchase intention of consumers of used golf clubs in the context of increasing C2C transactions and further explore the interaction effect of regulatory focus.” The problem with this aim is that it frames the current study as a replication study even though there is a degree of novelty in the findings. A shift of focus from the practical contribution to contribution to theory would probably give the study a better selling point.

Theory

This study builds on the regulatory focus theory and theorizing on the influence of trust on consumer behaviour. Thus, the study does not promise new major insights in buyer behaviour in consumer to consumer context. The first hypothesis pretty much repeats what has been observed and documented in C2C research during the past fifteen, or so, years. Hypothesis 2, in turn, has some merit, as I am not aware of studies testing similar interaction effect between regulatory focus and seller trust.

The authors should more carefully describe the regulatory focus theory: Promotion focus and prevention focus are not fixed traits. Rather, people display both promotion and prevention focus to some extent, and the balance depends on the context. 

Methods

Both data collection and data analyses are meritorious. The number of participants is sufficiently high to produce statistically significant differences in the data, and the exposure of participants to stimuli was randomized. Also, the analysis methods, independent samples t-test and two-way analysis of variance are suitable for the purposes of the study.

I can find room for only minor improvements. The most important, even if slightly cosmetic, improvement would be to use “<0.005” or similar expressions to report the p-values. I do understand that the authors are seeking to round the reported figures to two decimals, yet, I would prefer that they avoid reporting the p-values as “0.00”. Reporting of demographics seems a bit superfluous, as the key concepts are not related to them, and if there is some covariation, randomizing should take care of that.

Results

The authors report the results of their study as follows: “The results revealed a significant difference in purchase intention based on seller trust, with higher purchase intentions observed when seller trust was high compared to when it was low. Additionally, the interaction effect of regulatory focus was found to be significant in the impact of seller trust on golf club purchase intention.” 

The results are clearly reported, and they support the hypotheses, and hence align with the chosen theory base. I do not find technical problems with the reported results. While the results of the current study may be novel to sports science, this is partly a replication study when looked at from the point of view of most disciplines involved in e-commerce.

Contribution

The authors report their contribution as follows: “findings underscore the importance of seller trust in the context of an expanding market for online second-hand trading platforms.” The problem is that consumer trust has been the focus of numerous studies, and many of them document the effect of consumer trust on purchase intention, even in C2C contexts. As such, the findings and contribution are partly underwhelming. Then again, I find the reported interaction effect (H2) sufficiently interesting to merit publication.

Other

There are some difficult to read sentences that should be reformulated, such as the following sentence, starting on line 52: “In the past, consumers had to consider temporal and economic costs when selling products, often resorting to offline or online markets to find suitable sellers.” The word “middlemen” could be substituted to the last word (sellers) of the sentence to increase clarity.

Also, there are some inconsistencies in the use of some of the key concepts. For example, the following sentence, starting at line 94, suggests that sellers in the C2C context were not consumers: “C2C platforms, such as open markets, facilitate online transactions between numerous sellers and consumers.” In this case, it would be better to use the terms “seller” and “buyer”.

Reviewer 2 Report

Comments and Suggestions for Authors

Strengths

- Originality or innovativeness of the manuscript.

- Clear message and clarity of argumentation.

- The study was conducted in a coherent way.

- Manuscript structure.

- Well written and logical.

Weaknesses:

- The methodology approaches. Provide clear and better explanations for the

Research/Model design (is not explained).

- Discussion topic (are incipient and have a weak analytical-reflexive approach).

- Conclusions (the conclusions are very concise and brief, lacking fundamental analytical elements).

- Theoretical and managerial implications.

The manuscript adequately adheres to the mission and scope of the journal. The topic is timely, is relevant and current and relevant to scientific research. Clearly define the overall objectives. It is well written and logical. Presents relevant arguments to justify the research as a whole and the issue under investigation. The paper is explicit about its implications for research, practice and/or society. Although the study was conducted in a coherent way, it presents problems that compromise its contribution to the scientific community.

I summarize these problems in the following main points:

(1)      The manuscript does not comply with the standard journal references/bibliography. See Citation Guidelines on the journal official website.

(2)      The introduction does not include the necessary elements. It should be structured as follows: contextualization and relevance of the theme, objectives (general and specific), methodology, study contributions, article structure.

(3)      The paper does not undertake a critical approach in reviewing the literature. Absence of theoretical foundation and the most current references of the main topics (the first topic 1.1.1. has only 3 references from the last 5 years; the second topic 1.1.2. has no references from the last 5 years!). The literature review is not critical, reflective, and does not effectively show the state-of-the-art research in this area.

(4)      Non-systematization of study model, based on the hypothesis and the literature review.

(5)      The ethical procedures methodology is not explained.  Was the study presented to the university ethics committee? This is important to mention.

(6)   The analysed of data presented is not adequate, that is, the results discussion is very incipient. Does not present a true discussion of the results. The discussion of results is not critical, reflective, and does not effectively show the state-of-the-art research in this area. The absence of a critical and reflexive stance on the research results is noticeable. The discussion of the results must be more robust should be more reflective, incorporating more authors references to support the analyzes/results achieved.

(7)      The absence of theoretical and practical implications of the study is one of the weakest points of the paper. The theoretical and practical implications should be robust (for research, practice, and society), more consistent with the results obtained and being strengthened with the support of other studies.

(8)  The conclusions are incipient. The conclusions should be better discussed, based on contributions from some authors. Conclusions should also involve generalization of the comments of the authors regarding the issue, highlights, and research gaps. It should also present the next research steps.

Reviewer 3 Report

Comments and Suggestions for Authors

The Hypotheses were constructed upon an weak foundation, employing basic analysis, with unclear contributions and gaps in the research. Therefore, I recommend rejecting the research.

The writers should engage in an exhaustive exploration of existing literature, meticulously scrutinizing prior theories and models to inform the systematic development of their model. This process entails a comprehensive review of the pertinent literature, encompassing all relevant theories and models that have preceded the current study.

Arguably, the pivotal phase in any research endeavor is the construction of the model. This involves an in-depth examination of existing theories within the field, facilitating the identification of knowledge gaps. Subsequently, the authors must delineate their research's contribution and elucidate how it addresses the identified knowledge gap. Unfortunately, the current research falls short in explicating this critical aspect.

Round 2

Reviewer 2 Report

Comments and Suggestions for Authors

The manuscript has been sufficiently improved to warrant publication in Behavioral Sciences. However, my consideration “(4) non-systematization of study model, based on the hypothesis and the literature review” was not considered. I consider it important to include a figure with the analysis model and respective analysis at the beginning of the methodological section.

Author Response

Best regards,

Reviewer 3 Report

Comments and Suggestions for Authors

The revised manuscript appears to be inadequate as the authors have not adequately addressed the comments provided. Additionally, the paper lacks originality and does not make a significant contribution to the field.

Author Response

Best regards,
